# Free Riding without Dead Weight Losses

**Kwon-Sik Kim [1] and Seong-ho Jeong [2,\*]**

[1]   Korea Small Business Institute, Seoul 07074, Korea; kskim@kosbi.re.kr
[2]   Korea Public Finance Information Service, Seoul 04637, Korea
\*   Correspondence: jazzsh@kpfis.kr; Tel.: +82-10-7475-3724

**Abstract:** Traditional economic theory assumes that dead weight loss due to free riding on public goods is inevitable. This study demonstrates that free riding without dead weight losses can theoretically exist through Bowen's model. To this end, this study uses the consumer surplus analysis to present the conditions for free-riding that do not involve dead weight losses, as well as to demonstrate that policy choices that satisfy both the value of efficiency and equity in the supply of public goods are possible. This article formularizes the conditions under which such exceptional cases occur and examines what policy implications the presence of such conditions have in making decisions about the provision of public goods. The discussion of possibility and conditions for free-riding without dead weight losses is significant in that it suggests theoretical and policy implications for policies to raise equity as another important value, not just providing a solution to market failure.

**Keywords:** free-riding; dead weight losses; public good; market failure; non-excludable and non-rival

## 1. Introduction

It is a common conclusion of conventional economic analysis that the free-rider problem that is common with public goods leads to social inefficiency (i.e. dead weight loss). According to the approach of classical welfare economics, public goods, which are both non-excludable and non-rivalrous, cannot realize the optimum production where social welfare is maximized [1–3]. Such market failure in public goods is generally explained by the opportunistic behavior of free riders. Assuming that free-riding causes dead weight losses, the government's role in solving the free rider problem, one that has plagued the market for years, is important to the efficient use of public goods. Many theoretical discussions have been raised to resolve this. A few examples of such discussions have been suggested by Hirschman, such as the role of an activist [4], government provision [5], voluntary negotiated settlement [6], and settlement by assurance contract. On the other hand, if there are conditions in which free-riding does not result in social inefficiency, the government needs to take on a new role.

However, what if there are exceptional cases in which there is no dead weight loss even if free riding is allowed? If such cases do exist, it would be possible to achieve social equity without a loss of economic efficiency. Is this really possible, and if so, what are the political implications of this scenario?

Mainstream economics has been consistent in concluding that the free-rider problem causes dead weight loss in the provision of public goods [2,7,8]. The fundamental purpose of this study is to demonstrate, through the Bowen's model based upon the consumer surplus analysis approach, that free-riding without incurring social inefficiency theoretically exists. It also discusses the possibility of a policy choice that satisfies both efficiency and equity, two conflicting values in the supply of public goods, by presenting the conditions for free riding without the dead weight loss.

We explored the tendencies which can appear between the two desirable values, equity and efficiency. And the case of free-riding without dead weight loss can help promote social equity in public goods supply, without causing the hindrance of economic efficiency. Through this, a desirable social

value, equity can be achieved without the additional consumption of economic resources. And this, ultimately, can contribute to the "sustainability" of national finances and the whole community's economic resources.

This study is organized as follows: Chapter 2 is the literature review; Chapter 3 examines the conditions for the formation of free-riding that do not involve dead weight losses; Chapter 4 looks at the possibility of free-riding that exists without social inefficiency as well as the policy implications that can be selected for supplying public goods; and Chapter 5 concludes this study as a whole.

## 2. Literature Review

### 2.1. Theoretical Scope

Economists generally believe that the pareto optimality of resources in relation to public goods is incompatible with the underlying incentive of private ownership [3]. In this regard, many scholars have discussed alternatives to solve the problem of free riding in the supply of public goods. Hirschman [4] argues that individuals in fierce competition for survival in the private sector cannot afford to devote their resources to solving problems in the public domain, so they benefit from an activist who organizes collective action and solves public problems. Meanwhile, political entrepreneurs or leaders believe that individuals can address the problem of public goods by appealing to their own altruism. However, Friedman and other economists argue that the government should find a solution to the problem of public goods in other areas and not rely on calls for altruism. For example, Friedman insists on supporting a legal monopoly while excluding a technical monopoly (natural monopoly) since it might be more efficient for the government to provide services that cannot be provided directly by the private sector, although some among these can be provided more efficiently by the private sector as well [9].

Tabarrok [10] presents a solution through the assurance contract. When a certain quorum is reached in a manner that forms public goods through binding pledges, the public goods are supplied through a collection fee gathered from the participants, which in turn ultimately becomes a profit of the public good supplier. Coase [6] argues that if these are no transaction costs, it becomes easier for the beneficiaries of the public good to negotiate and resolve the public good problem by exchanging resources (Coase Theorem). In reality, transaction costs do exist, so the legal system or the government's role in reducing them plays a key role. In addition, to address the problem of free-riding inherent in the supply of public goods, the government can use taxes to secure funds and use government provision [10] or prevent free-riding entirely by making them mandatory through unfunded mandates [11]. Also, the government can subsidize the private sector to produce public goods [12]. All these classical theoretical discussions raise various solutions to the free-riding problem, but have the same major premise: free-riding harms pareto optimum of free-market.

### 2.2. Review of Recent Studies

Recent studies have primarily dealt with themes on the behavioral pattern and features through case studies or experimental methods. To begin with, some research has been conducted concerning the free-riding actors through case studies. Jordahl et al. [13] published the strategic behavior in the Swedish municipal amalgamation case, where each municipality had incentives to free-ride on their partners by increasing debt prior to amalgamation. Nordhaus [14] referred to international climate agreement (for example, Kyoto Protocol), in that a Climate Club with small trade penalties on non-participants can incur a large scale of stable coalition. Thielemann [15] argued, from the perspective of public goods theory, that some significant insights can be found in the effectiveness of EU refugee burden-sharing instruments.

In addition, there are many studies using the public goods game approach, mainly in economics. According to Fischbacher et al. [16], public goods experiments showed that many people contribute more to the public good than was previously thought by the pure self-interest assumption. But under certain conditions, free-riding still appears because of other-regarding preferences; for example,

frustrated attempts at kindness, learning the free-riding incentives, etc. Nielsen et al. [17] show by public goods game, that free-riders spend much time deliberating whether to break a social norm of conditional cooperation or not, so-called second thinking. Ellingsen et al. [18] maintained that in a contracting game of public goods supply, collective ownership enables more efficient consequences than individual asset ownership, due to contract negotiations.

Some sequential studies have been conducted on the free-riding behavior with other kinds of social experiment methodology. Bonroy et al. [19] argued the free-ride problem depends on the quality among members of a marketing cooperative, where the average quality provided by members of the cooperative results in a collective rent. Bucciol et al. [20] contended that peer monitoring could promote virtuous behavior when monetary incentives cannot be used to solve social dilemmas, in the case of free riding around household waste sorting. Iida et al. [21] argued that there is a general difference between a decision made as an individual and as a representative of a group in the context of a public good game, representatives contributed less than individuals when they could not communicate with their constituency. McDougal [22] studied the effect of social status on perceptions of, and reactions to, free riders.

As shown above, recent studies mainly focus on researching the behavioral pattern of free-riders through experiments or case study, and do not explore questions on the consequences of resource allocation in case free-riding.

### 2.3. The Distinction of This Study from Existing Studies

Existing discussions, whether classical or recent, mentioned above, are an attempt to find an exogenous solution (i.e. activist, administration provision, administration mandate, etc.) to the theoretical model, assuming that free riding of public goods is an inevitable phenomenon. This study verifies that there are circumstances in which the problem of free-riding can be solved endogenously through the unusual case inherent in a partial equilibrium model [1,23] on the supply of public goods. Additionally, this research poses the formularized conditions on that case and their policy implications. From these points, we can find the originality of this research. In the next chapter, we verify the existence and conditions of the case where free-riding does not incur dead weight loss in public goods supply.

## 3. Free Ride Cases without Market Failure

### 3.1. Free-Riders' Influence on the Public Goods

In case of market failure, the balance of the perfect market cannot maximize social welfare any longer. As for public goods, a kind of market failure, there will be losses of welfare (dead weight loss) because its non-rivalry and non-excludability makes it difficult for a market to form, and even if there is a market, there will be under-production due to the free-rider problem causing welfare losses, as production would not reach the optimal level.

The perfectly competitive market equilibrium achieved by the price mechanism maximizes social welfare. This can be demonstrated by showing that social welfare decreases when actual production is less than or exceeds the production $Q^*$ in perfect competition equilibrium. Figure 1 shows the situation in which social welfare loss occurs when the production in the actual market falls below the production $Q^*$, maximizing social welfare due to market failure. Production $Q^*$ in perfect competition equilibrium maximizes consumer and producer surplus, and consequently, social welfare. If actual production becomes $Q_1$, less than $Q^*$ due to market failure, consumer and producer surplus will decrease and there will be social welfare loss of $\triangle ABC$. This loss of welfare is referred to as "dead weight loss" [24].

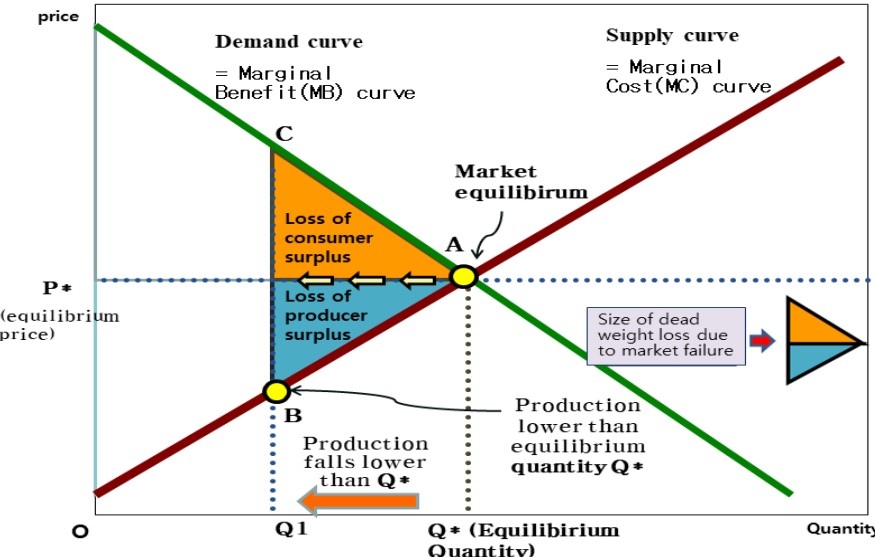

**Figure 1.** Social welfare loss due to market failure [24]. Note: * means equilibrium.

In the theory of finance, non-rivalry and non-excludability are cited as major characteristics of public goods as opposed to private goods [2,8,25]. Due to non-rivalry of the public goods, meaning use by one individual does not reduce its availability to others, these goods can be effectively consumed simultaneously by more than one person within the range of the benefits. Also, public goods have non-excludability, meaning individuals cannot be excluded from use for not paying for them, so the so-called free-rider problem occurs when individuals free ride on the contribution of others to enjoy benefits. Due to these characteristics, if the provision of public goods is at the hand of the free market, efficient allocation of resources cannot be achieved, and the more free-riders, the more distorted social decision making is, which causes inefficiency of resource allocation (dead weight loss) and, consequently, market failure. There have been a number of scholars who proved the existence of this free-ride phenomenon with various methodologies. Recently, the focus has been on confirming the possibility of free-riding through mathematical induction or social experiments [16,26–29].

Out of many theories and models which analyze market failure due to the free-rider problem, the four representative models are the prisoners' dilemma model by the same utility functions; prisoners' dilemma model by the different utility functions; public goods game social experiment based on prisoner's dilemma model; and analysis of consumer surplus of free-riding [30]. Consumer surplus analysis explains market failure due to free-riding well by introducing realistic assumptions that individuals have different utility functions, supply and demand of public goods are not fixed and divided, and marginal benefits vary depending on changes in supply and demand [30]. In this chapter, we analyze the case of free-riding without dead weight losses through consumer surplus analysis in contrast with prior research on free-riding with public goods.

### 3.2. Welfare Economics Perspective of Free-Riding

Private goods do not have non-rivalry or non-excludability as factors, and their consumption by one consumer prevents simultaneous consumption by other consumers, which means an individual cannot free ride on the consumption of these goods to enjoy their benefits. Therefore, market demand at a given price is the sum of consumption by individual consumers. Market demand would be the horizontal summation of all the market participants' individual demand while the market demand curve is represented by the horizontal summation of all the individual demand curves.

On the other hand, as presented in Figure 2, the relationship between individual demand curves and the market demand curve of public goods is different from that of private goods. As previously mentioned, the consumption of public goods are non-rivalrous, and it is possible for different consumers

to consume the same amount of public goods simultaneously. Therefore, the benefit of society as a whole from a certain quantity of public goods is the sum of the marginal benefits of individual consumers, so the social demand curve (social marginal benefit curve) is equal to the sum of individual demand curves. In Figure 2, the demand curve means the marginal benefit curve and the height of the demand curve measures the consumer's marginal willingness to pay (WTP) and the size of the marginal benefit.

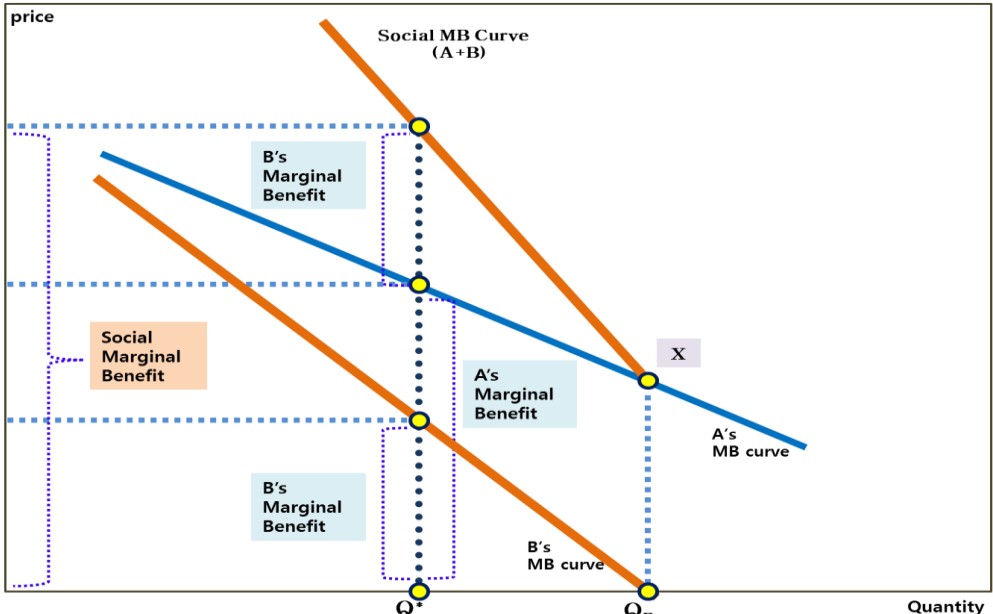

**Figure 2.** Individual demand curve and market demand curve of public goods [4,18]. Notes: 1. Public goods are non-rivalrous in consumption and the social marginal benefit curve is represented by the vertical summation of individual consumers' marginal benefit curves; 2. Social marginal benefit curve bends at point X if there is no free-rider; 3. If individuals express their benefits they are asked to bear the cost of producing public goods according to the marginal benefit they enjoy.

As suggested by Bowen [1], we intend to analyze the free-rider problem and dead weight loss through a partial equilibrium model [1,23]. In the situation where the market demand curve is obtained by the vertical summation of the individual consumers' marginal benefit curves, there is a gap between the social optimal production amount and the market equilibrium production realized in the actual market, and the welfare loss caused by this shows the market failure caused by free-riders. Let us consider a typical case where a dead weight loss is incurred by the free ride problem through a consumer surplus analysis model.

Assume there are two consumers; consumer A with relatively higher benefit to gain from a public good, meaning the height of the demand curve is higher than B, and consumer B with lower benefit from a public good, meaning the height of the demand curve is lower than A. For convenience's sake, A is referred to as a high-benefit consumer, while B is a low-benefit consumer. The marginal cost curve of the public goods production becomes the supply curve of public goods. In this case, the amount of dead weight loss can be obtained by comparing social optimum (F) and market equilibrium when there are free-riders (D, E). Let us examine the dead weight loss incurred by an individual free rider. The demand curve of high-benefit consumer A has a higher demand curve than that of consumer B (see Figure 3). In this case, actual market equilibrium, when B free rides on the contribution of A, and A pays for the provision of public goods alone, is point D where A's demand curve and supply curve meet, and social optimal equilibrium is F, where social demand curve, which reflects the true preferences of the two consumers, intersects the social supply curve. Therefore, when comparing point

F, where social welfare is maximized, and the actual market equilibrium D, it is evident that social welfare decreases by △ADF at point D.

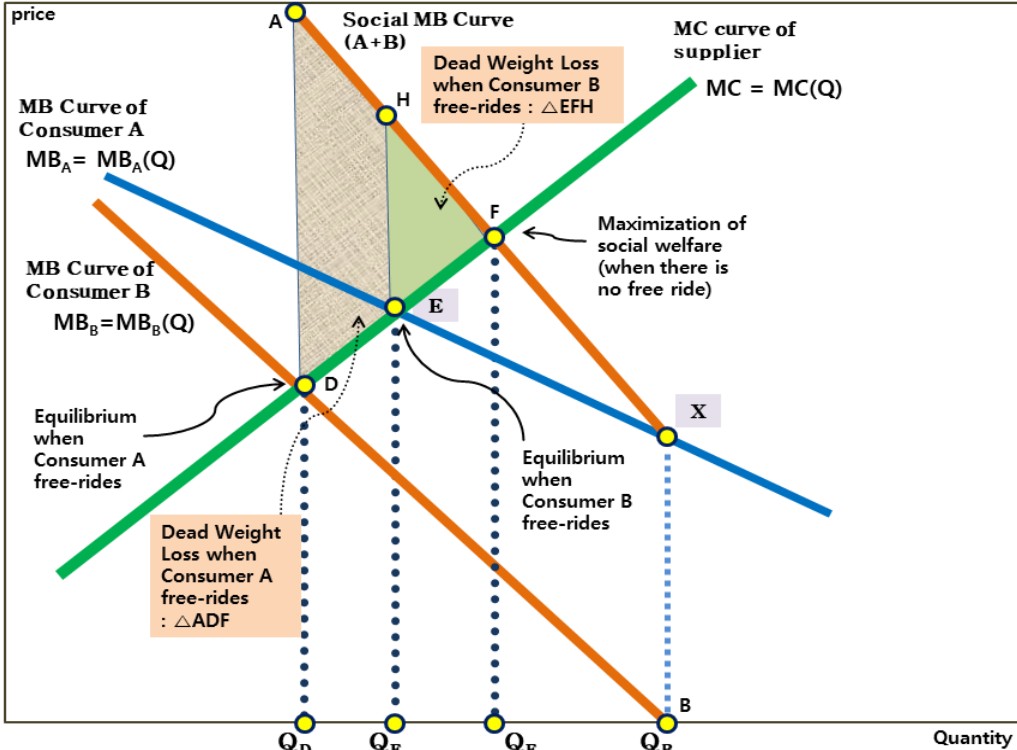

**Figure 3.** Market failure and dead weight loss by the free-rider. Notes: 1. Point B is production when Consumer B's benefit is zero; 2. Point D is Market equilibrium reached when A free-rides and B pays for all the cost; 3. Point E is Market equilibrium reached when B free-rides and A pays for all the cost; 4. Point F is Social welfare that is maximized when there is no free-rider.

On the other hand, when A free-rides on B's contribution and B pays for all the provision cost of the public good, actual market equilibrium is point E, where B's demand curve and supply curve meet. Thus, comparison of point F, where social welfare is maximized, and the actual market equilibrium E, shows that social welfare decreases △HEF at point E due to B's free ride. In this case, the dead weight loss incurred when a high-benefit consumer A free rides is greater than that of a low-benefit consumer B (△ADF > △HEF).

In short, classical welfare economics conclude that when a consumer free-rides, concealing his or her benefit and distorting social benefits, it leads to under-provision of public goods at the level lower than is required to maximize social welfare, incurring dead weight losses. This result justifies the argument of the traditional public economics that the government should intervene with policy measures to reach the optimal social welfare level (point F in the figure) to address dead weight losses associated with the free-rider problem [2,7,31]. However, further research is required to understand if free-riding always incurs dead weight losses, or, in other words, causes undesirable results in terms of economic efficiency. The next section examines the cases of free-riding, where there is no loss in economic efficiency or dead weight losses, from the perspective of welfare economics.

*3.3. Why Does Free-Riding Incur Dead Weight Losses?*

To begin with, why does a free rider generate dead weight losses? In this example, the reason is that the actual market equilibrium (point D and E) is positioned to the left of the bending point (point X) of the social marginal benefit curve. A closer look at Figure 4 reveals that in the section where there is a gap between the height of the social marginal benefit curve and that of Consumer A's marginal

benefit curve (left of point X), optimum balance, or maximum social welfare (point F) is reached at the intersection of the supply curve, and the provision of public goods differ between optimum balance F and actual market equilibrium D and E, generating dead weight losses. In other words, social welfare, the sum of consumer surplus and producer surplus, is maximized at optimum balance point F where the supply is $Q_F$, due to the free-rider problem, production is determined at actual market equilibrium D and E, where production $Q_D$ and $Q_E$ are lower than social optimal production quantity. Therefore, the size of social welfare through the market equilibrium (D, E) is smaller than the maximized social welfare at point F, by △ADF and △HEF, respectively (see Figure 3).

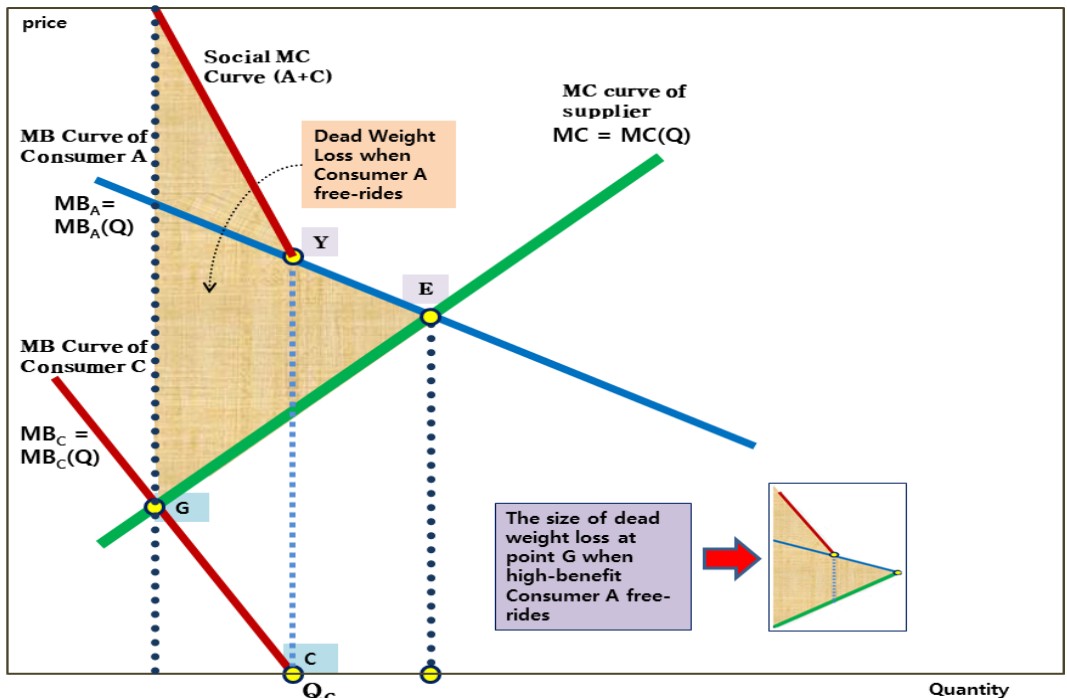

**Figure 4.** Free-riding without dead weight losses. Notes: 1. Point C is production when Consumer C's benefit is zero; 2. Point E is Market equilibrium reached when C free-rides and A pays for all the cost; 3. Point G is Market equilibrium reached when A free-rides and Consumer C pays for all the cost.

### 3.4. Possibility of Free Riding without Dead Weight Losses

Shin [12] suggests that there are cases in which free-riding does not lead to dead weight loss, unlike the conventional argument. Let us assume that low-benefit consumer C free-rides on the contribution of high-benefit consumer A, who pays for all the provisional costs of a public good (see Figure 4). In this case, the bending point of the social benefit curve (point Y) is located to the right of market equilibrium and is reached when there is free-riding (point E) and the location and shape of demand-supply curves for the public good becomes different than usual. When point E, where public goods supply curve intersects with high-benefit consumer A's demand curve, is located to the right of the bending point (Y), point E becomes the actual market equilibrium[1] and social optimum (welfare maximization), simultaneously, although low-benefit consumer C free-rides. Unlike the case explored in Figure 4, social welfare can be maximized at the market equilibrium where consumer C free-rides and only consumer A's demand is expressed. That is, as social welfare is maximized at the actual market equilibrium, there are no dead weight losses.

Figure 4 shows that the actual market equilibrium E, where demand curve for A—whose demand is expressed—and public goods supply curve meet when C free-rides, is located to the right of the bending point Y of the social marginal benefit curve. In this case, unlike in the case illustrated in Figure 3, the social benefit curve bends in the left of market equilibrium E and there is no gap between the actual market equilibrium and social optimum welfare, the two points are identical because

high-benefit consumer A's demand curve corresponds to the social demand curve to the right of the bending point Y.

On the other hand, in the case where low-benefit consumer C's demand is expressed and high-benefit consumer A free-rides, social welfare loss becomes larger than that of Figure 4. This is because when high-benefit consumers free-ride, there is a gap between the actual market equilibrium G and social optimum E, and production of public goods at point G does not reach that of the social optimum point E (See notes in Figure 5 for the size of dead weight loss).

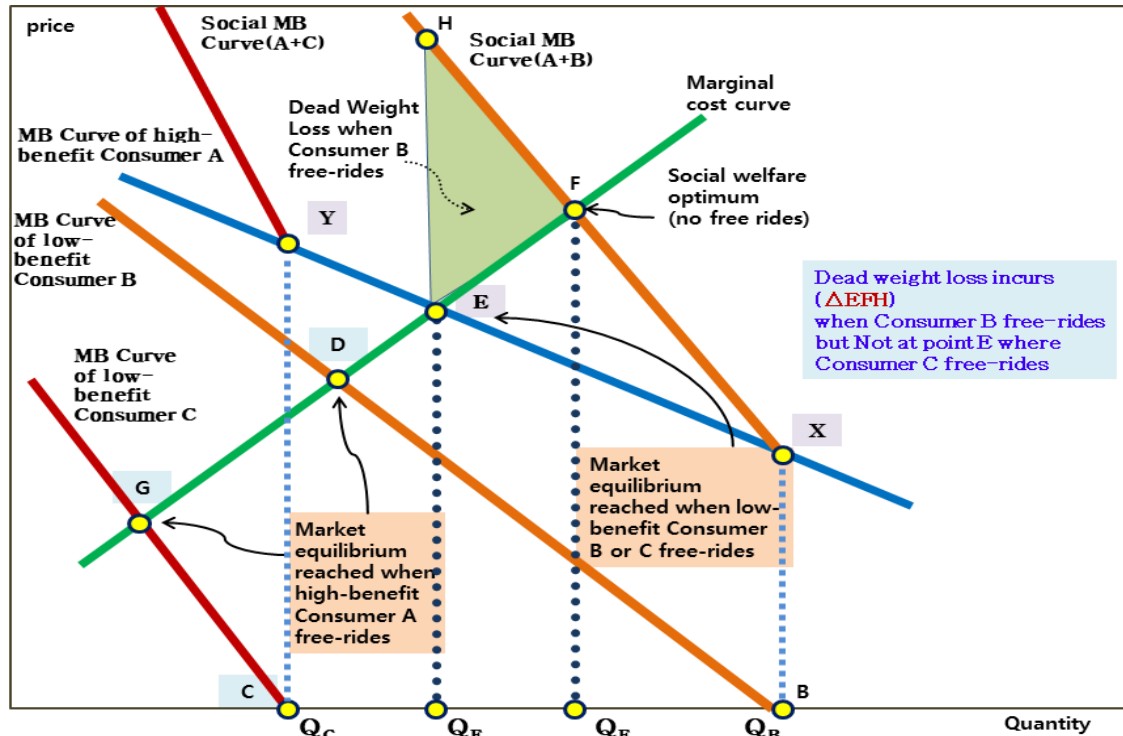

**Figure 5.** Relationship between free-riding and dead weight loss. Notes: 1. Point C is production when Consumer C's benefit is zero; 2. Point D is Market equilibrium reached when A free-rides and B pays for all the cost; 3. Point G is Market equilibrium reached when A free-rides and C pays for all the cost; 4. Point F is Social welfare is maximized and there is no free ride for A or B; 5. Point E is Low-benefit consumer B and C free-ride and A pays for all the cost. Dead weight loss incurs (△EFH) when Consumer B free-rides, but not at point E where Consumer C free-rides.

Figure 5 shows Figure 3, where free-riding incurs dead weight loss, and Figure 4, the case of free-riding without dead weight loss, simultaneously. If a high-benefit consumer is A and low-benefit consumer who free-rides is B, the social marginal benefit curve bends at point X. Therefore, actual market equilibrium is inconsistent with social optimum (E ≠ F), and the production of public goods at the actual market equilibrium $Q_E$ is smaller than social optimum production $Q_F$, creating dead weight loss of △EFH. On the other hand, when low-benefit consumer C free-rides, the social marginal benefit curve bends at point Y. Therefore, social welfare is at its maximum at the actual market equilibrium (E) where only A's demand curve is considered, because social marginal benefit curve is the same as high-benefit consumer A's demand curve on the right of point Y. In other words, if the low-benefit consumer who free-rides is C, rather than B, the $Q_E$ that is supplied from the actual market equilibrium becomes the production maximizing the social welfare immediately. Thus, dead weight loss is not incurred. On the other hand, if A free-rides on C, a larger dead weight loss is incurred than when A free-rides on B, which has already been explained in Figure 4.

In conclusion, whether dead weight loss will be created due to free-riding or not, depends on the relative position of the bending point on the social marginal benefit curve (demand curve for

the public good), whether it is located to the left or right of the actual market equilibrium E. Given this, what are the conditions for low-benefit consumers' free-riding not to incur social welfare loss? X-coordinate for bending points (Y, X) on the social marginal benefit curve means demand where low-benefit consumers' marginal benefit (or WTP) becomes 0 ($Q_C$, $Q_B$), F means social optimum where social marginal benefit curve intersects marginal cost curve and the social optimum production is $Q_F$. Under the circumstances, the conditions for low-benefit consumer C's free-riding not to incur dead weight loss, can be formulated as discussed below.

Let's assume the marginal benefit curve of consumer A enjoying a relatively high benefit from a certain public good, consumer C enjoying a relatively low benefit, and producer, as $MB_A = MB_A(Q)$, $MB_C = MB_C(Q)$, $MC = MC(Q)$, respectively. Market equilibrium E satisfies $MB_A(Q) = MC(Q)$. And where low-benefit consumer C free-rides without dead weight loss and high-benefit consumer A bears all the cost, the production at market equilibrium E is $Q_E$. Meanwhile, production where low-benefit consumer C's marginal benefit becomes 0 ($MB_C(Q) = 0$), is $Q_C$, which is also production at point Y where the social marginal benefit curve bends.

As mentioned above, when low-benefit group C free-rides on the contribution of high-benefit group A and there is no dead weight loss, point E corresponds to the actual market equilibrium and social optimum for maximization of social welfare and should be located to the right of bending point Y on the social benefit curve. This means the production that makes low-benefit consumer C's marginal benefit 0, needs to be the same as social optimum production or smaller, as represented as $Q_C \leq Q_E$. From the discussion above, it is evident that the incurring of dead weight loss is determined by the relative position of demand (X-intercept on the demand curve) making low-benefit consumer's WTP 0, which is the shape of the demand curve (marginal benefit curve) of low-benefit consumer, rather than supply side factors such as the slope or position of the supply curve (marginal cost curve) for the public good. And the Condition for free-riding without dead weight losses is as follows.

---

### Condition for free-riding that do not involve deadweight losses

$MB_A = MB_A(Q)$, $MB_C = MB_C(Q)$, $MC = MC(Q)$, where consumer A and C exist.

**Point E**: Production at market equilibrium, where low-benefit consumer C free-riding without dead weight loss

$Q_E$: high-benefit consumer A bears all the cost, which satisfies $MB_A(Q) = MC(Q)$

$Q_C$: production where low-benefit consumer C's marginal benefit becomes 0,
   i.e. $MB_C(Q) = 0$, Then the Condition where free-riding of low-benefit consumer does not cause dead weight loss, can be formulated as $[Q_C \leq Q_E]$

---

*3.5. Free-Rider Problem in the Relationship between Consumer Groups*

We assumed the existence of two consumers above and examined the special case where free-riding with public goods does not incur a dead weight loss. If this is applied to the relationship between the two groups of consumers, such an analysis will be able to provide practical implications for creating and implementing public policies. For the convenience of analysis, let us introduce the following assumptions.

3.5.1. Assumption 1: Divisibility of Consumer Groups

It is assumed that the consumers of a public good can be divided into two groups by the size of benefits from consuming the public good. In this case, the demand curve of each group becomes a collective demand curve that combines the marginal benefits of the individuals in the group.

### 3.5.2. Assumption 2: Linear Demand Function Moving Down and Right

It is assumed the demand function is a linear function moving down and to the right. Also, it is supposed that the demand curves of different groups do not intersect. In other words, the demand of the low-benefit group is always lower than that of the high-benefit group in all sections of demand for the public good. As before, groups which enjoy relatively high or low benefits, are referred to as the high-benefit group and the low-benefit group, respectively.

### 3.5.3. Assumption 3: Possibility of Allowing Free-Riding by Expressed Preferences and Policy Selection

It is assumed that the preferences of the two groups can be learned accurately. And intentional free-riding is impossible while free riding is possible only by government's public policy choice or social agreement.

Under these assumptions, previous discussions about free-riding between two consumers, can be applied to the relationship between two groups. As shown in Figure 5, the social marginal benefit curve bends at the point of public goods production where the low-benefit group C's marginal benefit (or WTP) becomes 0, and optimum balance E is positioned on the right of the bending point which intersects the marginal cost curve. Therefore, there is no gap between social optimum and the actual market equilibrium, and no dead weight loss incurred despite the free-riding of group C.

As demonstrated in Figure 6, let's assume the existence of two consumer groups, A and C. And the marginal benefit curve of consumer group A enjoying a relatively high benefit from a certain public good, consumer group C enjoys a relatively low benefit, and producer, as $MB_A = MB_A(Q)$, $MB_C = MB_C(Q)$, $MC = MC(Q)$, respectively.

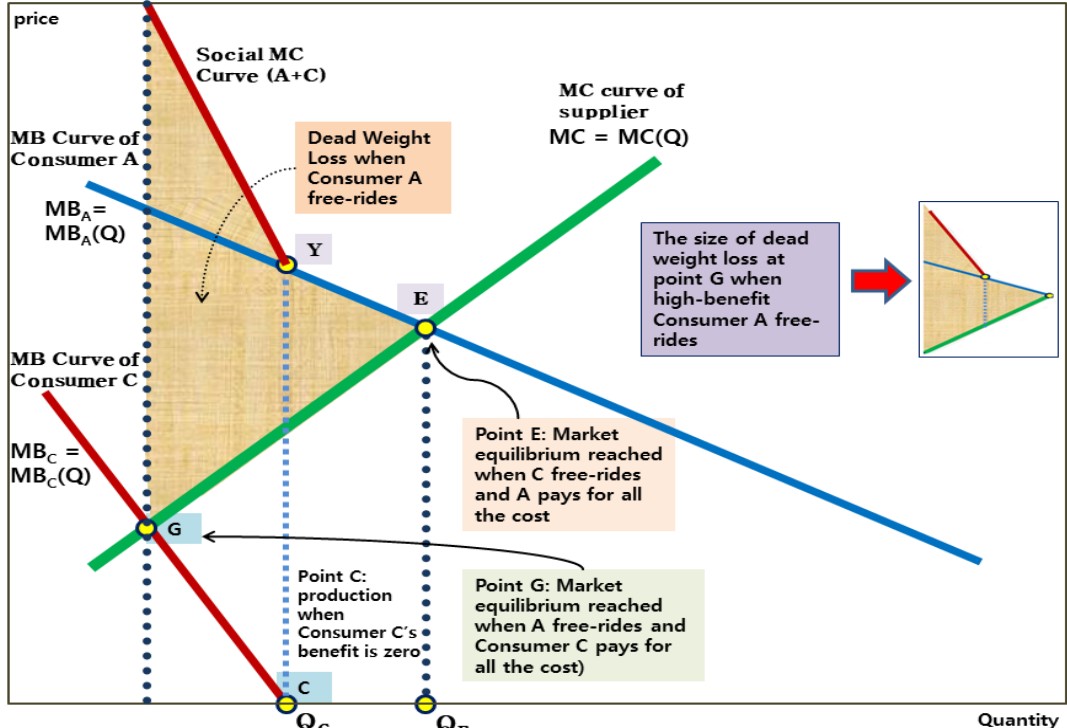

**Figure 6.** Cases of free-riding between groups with no dead weight loss. Notes: There is no dead weight loss at point E where low-benefit Group C free-rides and high-benefit Group A pays for all the cost. In contrast, there is considerably large dead weight loss incurred at point G where high-benefit Group A free-rides and low-benefit Group C pays for all the cost.

As explained in Section 3.4, at point E, where low-benefit consumer group C free-rides without dead weight loss, high-benefit consumer group A bears all the cost. At point $Q_E$, the condition, $MB_A(Q)$ = $MC(Q)$ is satisfied. Meanwhile, the production where low-benefit consumer group C's marginal benefit becomes 0 ($MB_C(Q) = 0$), is $Q_C$, which is also production at point Y where social marginal benefit curve is refracted. When low-benefit group C free-rides on the contribution of high-benefit group A and there is no dead weight loss, point E equals the actual market equilibrium (i.e. social optimum), and should be located to the right-side of bending point Y on the social benefit curve. As mentioned above, the production where low-benefit consumer group C's marginal benefit equal zero, meets the condition of $Q_C \leq Q_E$. And this condition [$Q_C \leq Q_E$] is the same as that of Section 3.4.

As demonstrated in Figure 5, whether dead weight loss will be created between groups, is also determined by the relative positions of bending points (X, Y) on the social marginal benefit curve and the actual market equilibrium E. Not only that, the difference in marginal benefits between the two groups is bigger in the case where there is no dead weight loss than where the low-benefit group free-rides and incurs a dead weight loss. In Figure 5, comparing low-benefit group C (no dead weight loss) and low-benefit group B, shows that the height difference in demand curves between A and C, is larger than that between A and B; indicating that whether dead weight loss will be created is determined by the relative positions of the demand curve of the low-benefit groups and resultant height difference between high-benefit group A and low-benefit group B, C. It is also clear that the difference in height of the demand curves between the two groups is much bigger when there is no dead weight loss. This means that the height difference in the demand curves between A and C, is much larger than that between A and B when the free-rider problem does not cause losses in economic efficiency. Given that the height of the demand curves of the two groups is more different when there is no dead weight loss, it is plausible to think that the higher the difference in benefits between the two groups with different income levels, the higher the possibility of free-riding not causing dead weight losses.

## 4. Discussion

As mentioned before, conventional economics assumes that free-riding incurs social inefficiency (dead weight loss) and several economists focused on the exogenous solutions based under this very assumption. As opposed to the conventional belief, this study proposes the possibility, formularized conditions and policy implications of the case on free-riding with no dead weight loss.

### 4.1. Possibility of Pursuing the Conflicting Values, Equity and Efficiency, at the Same Time

When a consumer free-rides on the contribution of others to use public goods, it leads to under-provision of public goods at the level lower than is required to maximize social welfare, incurring dead weight losses. However, there are cases where free-riding does not cause a dead weight loss. This was confirmed through the consumer surplus analysis above; if the production at the point where the benefit of a consumer group which enjoys relatively low benefit, becomes 0,[2] is the same as that of market equilibrium ($Q_E$), or smaller, social optimum balance is identical with the actual market equilibrium. Then what are the public policy implications of this situation where free-riders do not cause dead weight losses and can be justified in terms of economic efficiency? If free-riders do not inflict losses in social welfare, it is possible to produce and provide public goods to reach social optimum where all consumers can enjoy benefits without burdening every consumer with the production cost of the public goods. To be brief, it is possible to maximize economic efficiency of creating a maximum benefit at minimum cost.

On the other hand, there would be a problem of social equity or fairness for those consuming public goods without paying for them. Furthermore, we cannot rule out the possibility that some may conceal their preferences for public goods strategically causing dead weight losses, which can negatively influence the general public goods market.

Lakner and Milanovic [32] stated that the inequality of global income distribution gradually rose between 1988 and 2008, showing a pattern of polarization between the rich and the lower-income group.

Furthermore, this trend is becoming increasingly serious. Supply of public goods has a big impact on income distribution. Governments can spend income redistribution to strengthen social equity, but they can consider particular consumer groups by controlling the types and amounts of public goods that finance them. Given the worsening income polarization, it is all the more important to pursue measures to enhance social (distributional) equality between groups and classes by implementing public policies for fair distribution without hampering economic efficiency. It is the same for the supply of public goods. If consumers with lower income level are allowed to benefit from free riding on the cost burden of those with higher income, desirable allocation of resources from the perspective of social equity may be the result. Nevertheless, in terms of economic efficiency, free riding causes a dead weight loss and wasted resources. In this case, if the underprivileged is allowed to free ride in the name of social equity to justify the loss of economic efficiency, it will be very unlikely to be implemented as a policy because of many controversies and conflicts it will spark.

*4.2. Relationship between Income Levels and the Size of Benefits from Public Goods*

Then what determines the difference in marginal benefit between groups in relation to the conditions for not causing dead weight loss? This can be primarily seen as the standard to divide the consumers into two groups. If the two groups are divided by the income levels, government needs to identify the influences of the income level on marginal benefits gained from public goods. It is necessary to consider if the public goods' demand curve is higher, in other words, the benefit from public goods is bigger as the income level becomes higher.

A higher income level means higher benefit from public goods where those with higher income levels can get more help than those with lower incomes, from the services of public goods in protecting and managing their assets or income, or in consuming and enjoying them. On the other hand, having a lower income level means benefitting more from public goods, when those with lower income levels use their income to buy quality private goods as a substitute for public goods. For instance, there is public transportation service (although it is not a pure public good, but a quasi-public good charging fare) [30]. High-income earners are likely to benefit less from public transportation as they possess their own cars while low-income earners are likely to benefit relatively more from the public transportation service as they do not possess their own cars or do not use them as frequently. If it is true that those with higher income levels are more likely to enjoy greater benefits, it is more likely for those with lower income level to form a low-benefit group. This means more room for [i]considering those in the low-income bracket while achieving economic efficiency, in other words, allowing them to free ride when using public goods.

Table 1 presents the result of the analysis of actual data.[3] In 2016 Korean General Social Survey (KGSS), items asking the perception about the government performance by functions, were used. Government activities can be seen as police services provided by the government and the degree of performance was set as a proxy variable to represent the level of satisfaction with the public service. Government performance in the function of crime control (security) is shown in Table 1. The result shows that the higher the income level the higher the benefit from the police services and the coefficient is significant (0.016).

**Table 1.** Evaluation of government performance in police services.

| Independent Variables | Regression Coefficient | t-Value | Significance Probability (P > t) |
|---|---|---|---|
| Gender (male = 1) | 0.315 *** | 4.650 | 0.000 |
| Age | 0.009 *** | 3.390 | 0.001 |
| Marital status (married = 1) | 0.031 | 0.420 | 0.675 |
| Job (employed = 1) | −0.161 ** | −2.330 | 0.020 |
| Middle school graduate | 0.101 | 0.680 | 0.498 |
| High school graduate | −0.111 | −0.920 | 0.357 |
| College graduate | −0.272 * | −1.730 | 0.085 |
| University or higher | −0.076 | −0.580 | 0.562 |
| Self-reported income level | 0.049 ** | 2.410 | 0.016 |
| _cons | 2.003 | 9.250 | 0.000 |

Note: * $p < 0.1$, ** $p < 0.05$, *** $p < 0.01$.

The findings indicate that the income gap is likely to be greater as the gap between benefits is bigger, which means that low-benefit groups may free ride without causing market failure. Not only that, if the condition of [$Q_C \leq Q_E$] is satisfied and the low-benefit group has lower income than the high-benefit group, it is possible to reach a conclusion that allowing free-rider may help promote social equity without reducing economic efficiency.

*4.3. Contribution to the Academic Literature and Further Study*

This study expands the existing economic analysis model (Bowen's Model) to validate that there are cases for which conventional theories are unable to account (i.e. free-riding case without market failure). Furthermore, those exceptional cases and their conditions grant us some significant implications, which can help us with pursuing the two conflicting values, equity and efficiency simultaneously in public goods supply, considering that most of social policies require a huge amount of government expenditure.

On the other hand, this study constructed an analytical model under strict assumptions: divisibility of consumer groups, linear demand function, allowing free-riding by expressed preferences and policy selection. Thus, future study based on less strict assumptions is needed to derive generalized conclusions. Furthermore, specific analysis was not carried out to reflect differences in characteristics of individual public goods, such as whether they are public goods provided by the central government or those provided by local governments, and whether the source of resources is national or local taxes. Considering these problems, empirical studies of estimating demand for public goods through the use of contingent valuation method (CVM) are left for future studies.

**5. Conclusions**

Previously or currently, most studies and arguments adhere to the assumption that free-riding causes market failure in public goods supply. But this study took the possibility of free-riding without dead weight loss as the starting point for research. Based on our results, we can suggest a way of promoting social equity without hindering economic progress, setting this study apart from previous ones.

We demonstrated there are cases where free-riding in public goods does not cause a dead weight loss, unlike the conventional argument that free-riding leads to market failure by creating dead weight losses, and formulated the conditions. We confirmed that to prevent the incurring of dead weight loss with free-riders, the difference in the demand curve height between high-benefit and low-benefit groups should be big enough, and the production quantity where marginal benefit of low-benefit

group becomes 0, should be less than the social optimum. Additionally, it was found that in applying this conclusion in implementing public polices, an analytical framework can be provided to seek desirable public policy alternatives in terms of economic efficiency and fair distribution when the benefit difference as the criteria for dividing groups is consistent with the income level. In short, the higher the difference in benefits between the two groups with different income levels, the higher the possibility of free-riding not causing dead weight losses. In this case, we can allow the lower-income and lower benefit group to free-ride the other group, thus realizing the value of equity without market failure.

Social equality is one of the most important values for public policies to pursue. To achieve this goal, it should be sometimes inevitable to compromise economic efficiency to a certain degree. In this sense, the research here has a significant implication as it suggests the possibility of satisfying the two apparently conflicting values, efficiency and equity, simultaneously if there is a way to accomplish social equity without reducing economic efficiency.

Notes:

1. Since C free-rides without expressing his or her demand, the market equilibrium is formed at the intersection of the marginal benefit curve of public goods and A's demand curve.

2. This means quantity Qc, which corresponds to the bending point of the social marginal benefit curve.

3. For analysis, 2016 data from Korean General Social Survey by Sungkyunkwan University in South Korea was used. The survey data is freely available (accumulated data, 2003-2016).

**Author Contributions:** Conceptualization, K-S.K. and S.J.; Methodology, S.J.; Writing-Original Draft Preparation, K-S.K.; Writing-Review and Editing, S.J.; Validation, S.J.

**Funding:** This research received no external funding.

**Conflicts of Interest:** The authors declare no conflicts of interest.

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
