# Peer review of "Free Riding without Dead Weight Losses"

_sustainability, doi:10.3390/su11195168_

Round 1
Reviewer 1 Report
Comments are in the pdf attached.

Author Response
Dear editor and reviewers.
We are very grateful of anonymous reviews’ valuable comments to improve our manuscript. We did our best to develop our draft by reflecting the comments as follows. All revisions were written in red sentences in the revised manuscript. We look forward to hearing from your journal.
Response to Reviewer 1 Comments
Point 1 : Ln 19. how is it measured?
Response 1 : It is measured as global inequality by the Gini Index(Lakner, C.; Milanovic, B. 2013, page 9).
Point 2 : Ln 22. Figure 1 is not mentioned in the text at all. Correct this. It is hard to see the text in Figure 1. What is the source of data for Figure 1?
Response 2 : We removed Figure 1 of previous version, because it has little to do with our research.
Point 3 : Ln 26.Something is missing here?
Response 3 : We rewrote that phrase to a sentence(line 360).
It is the same for the supply of public goods.
Point 4 : Ln 36.Add sources
Response 4 : We added the source related to the approach of the classical welfare economics (line 28).
Point 5 : Ln 43. Add sources
Response 5 : We added the reference of that citation (line 40).
Point 6 : Ln 45. what are the linkages of this to the sustainability? This should be addressed.
Response 6 : We paid attention to the tendencies which can appear between the two desirable values, equity and efficiency. And the case of free-riding without dead weight loss can help promoting social equity in public goods supply, without causing the hindrance of economic efficiency. Through this, a desirable social value, equity can be achieved without unneccessary consumption of economic resources. And This, ultimately, can contribute the sustainability of the national finance and the whole community’s resources. To this end, we added some contents to the effect that free-riding without dead weight loss can be used in pursuing the conflicting values, equity and efficiency simultaneously, to the section 4.1 and conclusion (lines 352-366, 423-430).
Lakner and Milanovic [13] stated that the inequality of global income distribution was gradually rising between 1988 and 2008 showing a pattern ofpolarization between the rich and the lower-income group. Furthermore, thistrendisbecomingincreasinglyserious. Supply of public goods has a big impact on income distribution. Governments can spend income redistribution to strengthen social equity, but they can consider particular classes by controlling the types and amounts of public goods that finance them. Given the worsening income polarization, it is all the more important to pursue measures to enhance social (distributional) equality between groups and classes by implementing public policies for fair distribution without hampering economic efficiency. It is the same for the supply of public goods. If consumers with lower income level are allowed to benefit from free riding on the cost burden of those with higher income, it may result in desirable allocation of resources from the perspective of social equity. Nevertheless, in terms of economic efficiency, free riding causes a dead weight loss and wasted resources. In this case, if the underprivileged is allowed to free ride in the name of social equity to justify the loss of economic efficiency, it will be very unlikely to be implemented as a policy because of many controversies and conflicts it will spark. (line 352-366)
Given the worsening income polarization, it is safe to say that social equality is one of the most important values for public policies to pursue.To achieve this goal, it should be sometimes inevitable to compromise economic efficiency to a certain degree. In this sense, this research has a significant implication as it suggests the possibility of satisfying the two apparently conflicting values, ‘efficiency’ and ‘equity,’ simultaneously if there is a way to accomplish social equity without reducing, or minimizing, economic efficiency.This study assumes that the benefits are the same as the Lindahl’s model, and discussions on basic premises, such as preference expression, will proceed to further study. (line 423-430
Point 7 : Ln 45. What is the contribution and novelty? What has been done in past regarding this topic? This should be briefly said in the introduction.
Response 7 : We added the contribution and novelty of our research to the introduction section (lines 29-35, 40-49)
Assuming that free rides cause dead weight losses, the government’s role in solving the free-rider problem of public goods, a problem that has plagued the market for years, is very important. Many theoretical discussions have been raised to resolve this. A few examples of such discussions have been suggested by Hirschman such as the role of an Activist[12], Government Provision[24], Voluntary Negotiated Settlement[7], and Settlement by Assurance Contract. On the other hand, if there are conditions in which free rides do not result in social inefficiency, the government needs to take on a new role(lines 29-35)
The fundamental purpose of this study is to demonstrate, through the Lindahl’s model based upon the consumer surplus analysis approach, that free riding without incurring social inefficiency can theoretically exist. It also discusses the possibility of a policy choice that satisfies both efficiency and equity, two conflicting values in the supply of public goods, by presenting the conditions for free riding without the dead weight loss. The organization of this study is as follows: Chapter 2 will be literature review; Chapter 3 will be concerning the conditions for the formation of free rides that do not involve dead weight losses; Chapter 4 consists of the possibility of free rides that exist without social inefficiency as well as the policy implications that can be selected for supplying public goods; and Chapter 5 will conclude this study as a whole.(lines 40-49)
Point 8 : Ln 47. This is not literature review. Please rewrite this section into a true literature review in which sources are stated with their main results and conclusions.
Response 8 : We fully agree with your comment, so we made up for our literature review with adding to section 2.3(Theoretical Scope), a variety of academical arguments and opinions on the public goods and free-riding problem (lines 155-184).
Economists generally believe that the pareto optimality of resources in relation to public goods is incompatible with the underlying incentive of private ownership [11]. In this regard, many scholars have discussed alternatives to solve the problem of free riding in the supply of public goods. Hirschman [12] argues that individuals in fierce competition for survival in the private sector cannot afford to devote their resources to solving problems in the public domain, so they benefit from an activist (activist) who organizes collective action and solves public problems. Meanwhile, the political entrepreneurs or leaders believe that individuals can address the problem of public goods by appealing to their own altruism.However, Friedman and other economistsargue that the government should find a solution to the problem of public goods in other areas and not rely on the call for altruism. For example, Friedman insists in supporting legal monopoly while excluding technical monopoly (natural monopoly) since although it might be more efficient for the government to provide services that cannot be provided directly by the private sector, some among these can be provided more efficiently by the private sector as well[10].
Tabarrok [23] presents a solution through the assurance contract. When a certain quorum is reached in a manner that forms public goods through binding pledges, the public goods are supplied through a collection fee gathered from the participants, which in turn, ultimately become a profit of the public good supplier. Coase [7] argues if the transaction costs is ‘zero’, it makes it easier for the beneficiaries of the public good to negotiate and resolve the public good problem by exchanging resources (Coase Theorem). However, in reality, transaction costs do exist, so the legal system or the government’s role in reducing such transaction cost plays a key role.In addition, to address the problem of free rides inherent in the supply of public goods, the government can use taxes to secure funds and use government provision [23] orprevent free rides entirely by making them mandatory through unfunded mandates [8]. Also, the government can subsidize the private sector to produce public goods[21].
Existing discussions were an attempt to find an exogenous solution (i.e. activist, administration provisioning, administration mandate, etc.) to the theoretical model, assuming that free riding of public goods is an inevitable phenomenon. This study is to verify that there are circumstances in which the problem of free rides can be solved endogenously through the unusual case inherent in Lindahl's Model [3] on the supply of public goods, and the conditions under which such a situation can be established. (lines 155-184).
Point 9 : Ln 61. without below, it could be put above, thus not needed
Response 9 : We removed “below” (line 60).
Point 10 : Ln 67. what is MC and MB?
Response 10 : We replaced Figure 2 with Figure 1, which contains full name of MC and MB (line 57).
Point 11 : Ln 90. part = subsection 2.3. or?
Response 11 : We showed the subsection number explicitly (line 88).
Point 12 : Ln 91. Is this your contribution or existing research?
Response 12 : We added the contribution of our research distinct from existing researches (line 88-89).
In the next chapter 3, we will analyze the case of free rides without dead weight losses through consumer surplus analysis in contrast with prior research on free rides with public goods. (line 88-89).
Point 13 : Ln 182. extend this part of the figure.
Response 13 : We replaced Figure 5 and Figure 7 with Figure 4 and Figure 6 (lines 213, 302).
Point 14 : Ln 231. Such approach should be emphasized more in the paper. Namely, the theoretical modelling should be more straightforward: assumptions, mathematical formulae and relations and then the extensive description as given for previous Figures.
Response 14 : We showed the mathematical formulation of free-rides without dead weight loss in the box at section 3.3. (line 320)
Point 15 : Ln 254. some of the assumptions are restrictive. This is, of course, a starting point to conduct the analysis. However, assumption 2 is also a drawback of this model, i.e. problem. Shortfalls of the study should be added in the conclusion part and related to future work.
Response 15 : We are fully agreed your comments, and revised section 3.2. and conclusion according as your comments (lines 406-407, 428-430)
This study constructed an analytical model under strict assumptions. Thus, future study based on less strict assumptions is needed to derive generalized conclusions.(lines 406-407)
This study assumes that the benefits are the same as the Lindahl’s model, and discussions on basic premises, such as preference expression, will proceed to further study.(lines 428-430)
Point 16 : Ln 353. the rigorous analytical framework is missing in this research. This should be corrected, as I stated previously
Response 16 : We removed “rigorous” (line 420).
Point 17 : Ln 369. “was” should be added before “used”
Response 17 : We added “was” to notes 3 (line 438).
Reviewer 2 Report
The introduction of the article is quite austere and the reader is very poorly informed about the issue. Figure 1 is not mentioned and explained in the text and has no source. Likewise, the other images in the article do not have a source.
In the introductory part of the article, at the end of the Introduction chapter, there is no description of the following sections of the article and a brief summary of their content. At the end of each section of the article (lines 89-91 and also 152-154) is listed, which follows in the next section. However, this should be summarized at the beginning of the article. It is also necessary to add what is the main objective of the article, which is its focus and its contribution, where the authors see the scientific gap and how their article is situated in the current state of the issue. These aspects of the article should also be summarized in the introduction.
Chapter 2 is entitled Literature review, but it is not intended as a review of current studies by authors dealing with this issue and does not mention their current results. Subchapter 2.1, as drafted, does not belong to Literature review and does not contain any references.
The sources are not numbered in the correct order, the source number 1 is immediately followed by the number 5 (line 71). It is necessary to correct the order of references in the text of the article and also their citations.
Section 4 is called Discussion, but actually presents the results of the study. It would be useful to rename this section and add Discussion to the article as it is usual in scientific articles. In fact, there is a complete lack of a Discussion section, where the results of this study shoud be compared with similar studies (these should be added in the Literature review) and would be given the positioning of this study among others dedicated to this issue. There is also a lack of highlighting the weaknesses and limits of this study and the possibility of its further direction.
References are not adjusted to the prescribed format and must be edited. It would be highly advisable to incorporate a number of more recent references from the last 5 years to highlight the timeliness of the issue.
To summarize, the issue is interesting and up to date, but meeting the requirements for writing scientific articles is very weak.
Author Response
Response to Reviewer 2 Comments
Point 1: The introduction of the article is quite austere and the reader is very poorly informed about the issue. Figure 1 is not mentioned and explained in the text and has no source. Likewise, the other images in the article do not have a source.
Response 1 : We removed Figure 1 because it has little to do with our research. And We totally rewrote the introduction part in accordance with the whole contents of our research (lines 24-49).
It is the common conclusion of conventional economic analysis that the free-rider problem that is common with public goods leads to social inefficiency (i.e. dead weight loss).According to the approach of the classical welfare economics, public goods, which are both non-excludable and non-rivalrous, cannot realize the optimum production where social welfareis maximized[3,6,11]. Such market failure in public goodsis generally explained by the opportunistic behavior of free-riders. Assuming that free rides cause dead weight losses, the government’s role in solving the free-rider problem of public goods, a problem that has plagued the market for years, is very important. Many theoretical discussions have been raised to resolve this. A few examples of such discussions have been suggested by Hirschman such as the role of an Activist[12], Government Provision[24], Voluntary Negotiated Settlement[7], and Settlement by Assurance Contract. On the other hand, if there are conditions in which free rides do not result in social inefficiency, the government needs to take on a new role.
However, what if there are exceptional cases with no dead weight loss even if free riding is allowed? If such a case does exist, it would be possible to achieve social equity without a loss of economic efficiency. Is it really possible? If possible, what is the political implication of that case?
Mainstream economics has been consistent in concluding that the free-rider problem causes dead weight loss in the provision of public goods[1,6,15].The fundamental purposeof this study is to demonstrate, through the Lindahl’s model based upon the consumer surplus analysis approach, that free riding withoutincurring social inefficiency can theoretically exist. It also discusses the possibility of a policy choice that satisfies both efficiency and equity, two conflicting values in the supply of public goods, by presenting the conditions for free riding without the dead weight loss.The organizationof this study is as follows: Chapter 2 will be literature review; Chapter 3 will be concerning the conditions for the formation of free rides that do not involve dead weight losses; Chapter 4 consists of the possibility of free rides that exist without social inefficiency as well as the policy implications that can be selected for supplying public goods; and Chapter 5 will conclude this study as a whole. (line 24-49).
Point 2 : In the introductory part of the article, at the end of the Introduction chapter, there is no description of the following sections of the article and a brief summary of their content. At the end of each section of the article (lines 89-91 and also 152-154) is listed, which follows in the next section. However, this should be summarized at the beginning of the article. It is also necessary to add what is the main objective of the article, which is its focus and its contribution, where the authors see the scientific gap and how their article is situated in the current state of the issue. These aspects of the article should also be summarized in the introduction.
Response 2 : We added the focus, implications and contributions of our research to the introduction section (lines 36-44). And we showed the contents of each sections in the introductory part (lines 45-49).
However, what if there are exceptional cases with no dead weight loss even if free riding is allowed? If such a case does exist, it would be possible to achieve social equity without a loss of economic efficiency. Is it really possible? If possible, what is the political implication of that case?
Mainstream economics has been consistent in concluding that the free-rider problem causes dead weight loss in the provision of public goods[1,6,15].The fundamental purposeof this study is to demonstrate, through the Lindahl’s model based upon the consumer surplus analysis approach, that free riding withoutincurring social inefficiency can theoretically exist. It also discusses the possibility of a policy choice that satisfies both efficiency and equity, two conflicting values in the supply of public goods, by presenting the conditions for free riding without the dead weight loss. (lines 36-44)
The organizationof this study is as follows: Chapter 2 will be literature review; Chapter 3 will be concerning the conditions for the formation of free rides that do not involve dead weight losses; Chapter 4 consists of the possibility of free rides that exist without social inefficiency as well as the policy implications that can be selected for supplying public goods; and Chapter 5 will conclude this study as a whole. (lines 45-49)
Point 3 : Chapter 2 is entitled Literature review, but it is not intended as a review of current studies by authors dealing with this issue and does not mention their current results. Subchapter 2.1, as drafted, does not belong to Literature review and does not contain any references.
Response 3 : We are fully agree with your comment, so we made up for our literature review with adding to subchapter 2.3(Theoretical Scope), a variety of academical arguments and opinions on the public goods and free-riding problem (lines 155-184).
Economists generally believe that the pareto optimality of resources in relation to public goods is incompatible with the underlying incentive of private ownership [11]. In this regard, many scholars have discussed alternatives to solve the problem of free riding in the supply of public goods. Hirschman [12] argues that individuals in fierce competition for survival in the private sector cannot afford to devote their resources to solving problems in the public domain, so they benefit from an activist (activist) who organizes collective action and solves public problems. Meanwhile, the political entrepreneurs or leaders believe that individuals can address the problem of public goods by appealing to their own altruism.However, Friedman and other economistsargue that the government should find a solution to the problem of public goods in other areas and not rely on the call for altruism. For example, Friedman insists in supporting legal monopoly while excluding technical monopoly (natural monopoly) since although it might be more efficient for the government to provide services that cannot be provided directly by the private sector, some among these can be provided more efficiently by the private sector as well[10].
Tabarrok [23] presents a solution through the assurance contract. When a certain quorum is reached in a manner that forms public goods through binding pledges, the public goods are supplied through a collection fee gathered from the participants, which in turn, ultimately become a profit of the public good supplier. Coase [7] argues if the transaction costs is ‘zero’, it makes it easier for the beneficiaries of the public good to negotiate and resolve the public good problem by exchanging resources (Coase Theorem). However, in reality, transaction costs do exist, so the legal system or the government’s role in reducing such transaction cost plays a key role.In addition, to address the problem of free rides inherent in the supply of public goods, the government can use taxes to secure funds and use government provision [23] orprevent free rides entirely by making them mandatory through unfunded mandates [8]. Also, the government can subsidize the private sector to produce public goods[21].
Existing discussions were an attempt to find an exogenous solution (i.e. activist, administration provisioning, administration mandate, etc.) to the theoretical model, assuming that free riding of public goods is an inevitable phenomenon. This study is to verify that there are circumstances in which the problem of free rides can be solved endogenously through the unusual case inherent in Lindahl's Model [3] on the supply of public goods, and the conditions under which such a situation can be established.
Point 4 : The sources are not numbered in the correct order, the source number 1 is immediately followed by the number 5 (line 71). It is necessary to correct the order of references in the text of the article and also their citations.
Response 4 : We corrected the order of references and their citations in the whole text of the article (lines 440-480).
Banzhaf, H. S. The Market for Local Public Goods. Case Western Reserve Law Review. 2014, 64, 1441-1480. Baron, D. Morally Motivated Self-regulation. American Economic Review. 2010, 100, 1299– Bowen, H. R. Toward Social Economy. New York: Rinchart, 1948. Bracht, J.; Figuieres, C.; Ratto, M. Relative Performance of Two Simple Incentive Mechanisms in a Public Goods Experiment. Journal of Public Economics. 2008, 92, 54– Burlando, R.; Guala, F. Heterogeneous Agents in Public Goods Experiments. Experimental Economics. 2005, 8, 35– Case, K. E. Musgrave’s Vision of the Public Sector: The Complex Relationship between Individual, Society and State in Public Good Theory. Journal of Economics & Finance. 2008, 32, 348-355. Coase, R. The Problem of Social Cost. Journal of Law and Economics. 1960, 3, 1–44 Dilger, R. J., and Richard S. Beth. Unfunded Mandate Reform Act: History, Impact, and Issues. 2013. Fischbacher, U.; Gächter, S. Social Preferences, Beliefs, and the Dynamics of Free Riding in Public Goods Experiments. American Economic Review. 2010, 100, 541– Friedman, M.; Rose, D. Capitalism and Freedom, University of Chicago Press, 1982. Groves, T.; Ledyard, J. Optimal Allocation of Public Goods: A Solution to the "Free-Rider" Problem. Econometrica. 1977, 45, 783-809 Hirschman, Albert. O. Shifting involvements: Private interest and public action. Princeton, N.J.: Princeton University Press.2002. Lakner, C.; Milanovic, B. Global Income Distribution from the Fall of the Berlin Wall to the Great Recession. Policy Research Working Paper 6719. 2013. Mankiw, N. G. Principles of Economics. 6th ed, Cengage Learning, 2008. Musgrave, R. A. The Theory of Public Finance: A Study in Public Economy. New York: McGraw-Hill, 1959. Nikiforakis, N.; Normann, H. A Comparative Statics Analysis of Punishment in Public-Good Experiments. Experimental Economics. 2008, 11, 358– Reuben, E.; Riedl, A. Public goods provision and sanctioning in privileged groups. Journal of Conflict Resolution. 2009, 53, 1, 72-93. Rodrigues, Joã Where to Draw the Line between the State and Markets? Institutionalist Elements in Hayek’s Neoliberal Political Economy. Journal of Economic Issues. 2012, 46, 1007-1033. Salamon, Lester M.; Michael S. Lund. Beyond Privatization: The Tools of Government Action. Washington, D.C.: Urban Institute, 1989 Samuelson, P. A. The Pure Theory of Public Expenditure. The Review of Economics and Statistics. 1954, 36, 387– Shin, H. G. Consumer Surplus Analysis for the ‘Tragedy of Free-riding?’. Korean Policy Studies Review. 2016a, 25, 221-236. Shin, H. G ‘Prisoner’s Dilemma’ ‘Consumer Surplus’ As an Analysis of Free-riding. The Korean Association for Policy Development. 2016b, 16, 1-23. Tabarrok, A. The private provision of public goods via dominant assurance contracts. Public Choice.1998, 96, 345– Thompson, D. The Proper Role of Government: Considering Public Goods and Private Goods. The Pennsylvania State University, 2015.
Point 5 : Section 4 is called Discussion, but actually presents the results of the study. It would be useful to rename this section and add Discussion to the article as it is usual in scientific articles. In fact, there is a complete lack of a Discussion section, where the results of this study should be compared with similar studies (these should be added in the Literature review) and would be given the positioning of this study among others dedicated to this issue. There is also a lack of highlighting the weaknesses and limits of this study and the possibility of its further direction.
Response 5 : We revised section 4 according as your comments (lines 329-334, 352-366, 403-412).
As mentioned before, conventionaleconomics assumes that free riding incurs social inefficiency (dead weight loss) and severaleconomistsfocused on the exogenous solutions based under this very assumption. As opposed to the conventional belief, this study proposes the possibility of free riding with no dead weight loss and affirmsthat the level of income of the beneficiaries may vary. It also shows the acceptance of dead weight loss can achieve equality without compromising economic efficiency. (lines 329-334).
Lakner and Milanovic [13] stated that the inequality of global income distribution was gradually rising between 1988 and 2008 showing a pattern ofpolarization between the rich and the lower-income group. Furthermore, thistrendisbecomingincreasinglyserious. Supply of public goods has a big impact on income distribution. Governments can spend income redistribution to strengthen social equity, but they can consider particular classes by controlling the types and amounts of public goods that finance them. Given the worsening income polarization, it is all the more important to pursue measures to enhance social (distributional) equality between groups and classes by implementing public policies for fair distribution without hampering economic efficiency. It is the same for the supply of public goods.If consumers with lower income level are allowed to benefit from free riding on the cost burden of those with higher income, it may result in desirable allocation of resources from the perspective of social equity. Nevertheless, in terms of economic efficiency, free riding causes a dead weight loss and wasted resources. In this case, if the underprivileged is allowed to free ride in the name of social equity to justify the loss of economic efficiency, it will be very unlikely to be implemented as a policy because of many controversies and conflicts it will spark. (lines 352-366).
This study expands the existing economic analysis model to validate that there are cases for which conventional theories are unable to account. Thus, these theories fall short of proving that theoretical constructs derived under certain assumptions can indeed be found in the real world.This study constructed an analytical model under strict assumptions. Thus, future study based on less strict assumptions is needed to derive generalized conclusions.Furthermore, specific analysis was not carried out to reflect differences in characteristics of individual public goods, such as whether they are public goods provided by the central government or those provided by local governments, and whether the source of resources is national or local taxes. Considering these problems, empirical studies of estimating demand for public goods through the use of contingent valuation method (CVM) are left as a task for future studies. (lines 403-412).
Point 6 : References are not adjusted to the prescribed format and must be edited. It would be highly advisable to incorporate a number of more recent references from the last 5 years to highlight the timeliness of the issue.
Response 6 : Our research handles mainly the classical discussions of economic theories. Therefore most literatures on that subjects are classical, old-established articles. Furthermore current articles about public goods and free-riding are mostly based on those classical literatures. We corrected the format of references and added several articles including relatively recent established, as references (lines 440-480).
Point 7 : To summarize, the issue is interesting and up to date, but meeting the requirements for writing scientific articles is very weak.
Response 7 : We appreciate your comments, and they urged us to pay more attention to requirements for writing scientific articles. In addition to revise our manuscript, we will keep your valuable advices in our mind from now on for writing better scientific articles. Again, we are specially thankful for your salutary comments.
Reviewer 3 Report
The abstract need improvement. Authors need to describe what was the aim of their research and provide clear methods and results.
The introduction requires full rewriting. The authors need to provide the research problem and what is the gap they are going to close. There is a need for better introduction of current situation in the field of the research on free-riding problem. Aim, research question are needed.
Literature review requires improvement. It can't be based on such small amount of the positions especially in case when the journal is oriented on the international audience.
Author Response
Response to Reviewer 3 Comments
Point 1 : The abstract need improvement. Authors need to describe what was the aim of their research and provide clear methods and results.
Response 1 : We added the aim, methods and result of our research to the abstract (lines 9-19).
The traditional economic theory assumes that dead weight loss due to free rides on publics are inevitable. This study shows that free rides without dead weight losses can theoretically exist through the Lindahl’s model. To this end, this study uses the consumer surplus analysis to present the conditions for free rides that do not involve dead weight losses, as well as to demonstrate that policy choices that satisfy both the value of efficiency and equity in the supply of public goods are possible. This article aims to formularize the conditions under which such exceptional cases occur and to make an in-depth examination of what policy implications the presence of such conditions have in making decisions about the provision of public goods. The discussion of possibility and conditions for free rides without dead weight losses is significant in that it suggests theoretical and policy implications for the possibility of public policies to raise equity as another important value, not just providing a cure for market failure. (lines 9-19).
Point 2 : The introduction requires full rewriting. The authors need to provide the research problem and what is the gap they are going to close. There is a need for better introduction of current situation in the field of the research on free-riding problem. Aim, research questions are needed.
Response 2 : We added the aim, research question and introduction of current situation in the field of the research on free-riding problem in the introduction section (lines 24-49). And the trend of the research on free-riding problem is showed more specifically in section 2.3. (lines 155-184).
It is the common conclusion of conventional economic analysis that the free-rider problem that is common with public goods leads to social inefficiency (i.e. dead weight loss).According to the approach of the classical welfare economics, public goods, which are both non-excludable and non-rivalrous, cannot realize the optimum production where social welfareis maximized[3,6,11]. Such market failure in public goodsis generally explained by the opportunistic behavior of free-riders. Assuming that free rides cause dead weight losses, the government’s role in solving the free-rider problem of public goods, a problem that has plagued the market for years, is very important. Many theoretical discussions have been raised to resolve this. A few examples of such discussions have been suggested by Hirschman such as the role of an Activist[12], Government Provision[24], Voluntary Negotiated Settlement[7], and Settlement by Assurance Contract. On the other hand, if there are conditions in which free rides do not result in social inefficiency, the government needs to take on a new role.
However, what if there are exceptional cases with no dead weight loss even if free riding is allowed? If such a case does exist, it would be possible to achieve social equity without a loss of economic efficiency. Is it really possible? If possible, what is the political implication of that case?
Mainstream economics has been consistent in concluding that the free-rider problem causes dead weight loss in the provision of public goods[1,6,15].The fundamental purposeof this study is to demonstrate, through the Lindahl’s model based upon the consumer surplus analysis approach, that free riding withoutincurring social inefficiency can theoretically exist. It also discusses the possibility of a policy choice that satisfies both efficiency and equity, two conflicting values in the supply of public goods, by presenting the conditions for free riding without the dead weight loss.The organizationof this study is as follows: Chapter 2 will be literature review; Chapter 3 will be concerning the conditions for the formation of free rides that do not involve dead weight losses; Chapter 4 consists of the possibility of free rides that exist without social inefficiency as well as the policy implications that can be selected for supplying public goods; and Chapter 5 will conclude this study as a whole. (line 24-49)
Economists generally believe that the pareto optimality of resources in relation to public goods is incompatible with the underlying incentive of private ownership [11]. In this regard, many scholars have discussed alternatives to solve the problem of free riding in the supply of public goods. Hirschman [12] argues that individuals in fierce competition for survival in the private sector cannot afford to devote their resources to solving problems in the public domain, so they benefit from an activist (activist) who organizes collective action and solves public problems. Meanwhile, the political entrepreneurs or leaders believe that individuals can address the problem of public goods by appealing to their own altruism.However, Friedman and other economistsargue that the government should find a solution to the problem of public goods in other areas and not rely on the call for altruism. For example, Friedman insists in supporting legal monopoly while excluding technical monopoly (natural monopoly) since although it might be more efficient for the government to provide services that cannot be provided directly by the private sector, some among these can be provided more efficiently by the private sector as well[10].
Tabarrok [23] presents a solution through the assurance contract. When a certain quorum is reached in a manner that forms public goods through binding pledges, the public goods are supplied through a collection fee gathered from the participants, which in turn, ultimately become a profit of the public good supplier. Coase [7] argues if the transaction costs is ‘zero’, it makes it easier for the beneficiaries of the public good to negotiate and resolve the public good problem by exchanging resources (Coase Theorem). However, in reality, transaction costs do exist, so the legal system or the government’s role in reducing such transaction cost plays a key role.In addition, to address the problem of free rides inherent in the supply of public goods, the government can use taxes to secure funds and use government provision [23] orprevent free rides entirely by making them mandatory through unfunded mandates [8]. Also, the government can subsidize the private sector to produce public goods[21].
Existing discussions were an attempt to find an exogenous solution (i.e. activist, administration provisioning, administration mandate, etc.) to the theoretical model, assuming that free riding of public goods is an inevitable phenomenon. This study is to verify that there are circumstances in which the problem of free rides can be solved endogenously through the unusual case inherent in Lindahl's Model [3] on the supply of public goods, and the conditions under which such a situation can be established. (lines 155-184).
Point 3 : Literature review requires improvement. It can't be based on such small amount of the positions especially in case when the journal is oriented on the international audience.
Response 3 : We fully agree with your comment, so we made up for our literature review with adding a variety of academical arguments and opinions on the public goods and free-riding problem (lines 155-184).
Round 2
Reviewer 1 Report
Sustainability is not addressed in introduction (see my previous comment at end of intro).
Figure 2 - you added reference [3] - I did not find this graph in that book, which page is this on?
By emphasizing the mathematical formulae (page 8/12 comment in first draft) - I did not mean that you put it in a frame. I meant that you should provide more formulae throughout the paper and comments.
Literature review, and overall literature is very dated. How come that there aren't new findings in the last couple of years?
Discussion is a summary now. Rather, it should be focused on consequences of the findings, policy implications, theory, comparing results with previous findings, etc.
Author Response
Dear reviewers.
We are very grateful of anonymous reviews’ valuable comments to improve our manuscript. We did our best to develop our draft by reflecting the comments as follows. All revisions were written in Green sentences in the revised manuscript. We also requested a professional native speaker to edit our manuscript as suggested by the reviewer. We look forward to hearing from your journal.
Response to Reviewer 1 Comments
Point 1 : Sustainability is not addressed in introduction (see my previous comment at end of intro).
Response 1 : We added the explanation about “sustainability” to introduction and section 4.1.
Point 2 : Figure 2 - you added reference [3] - I did not find this graph in that book, which page is this on?
Response 2 : The account of Bowen’s model is presented in reference [4] and [18].
Point 3 : By emphasizing the mathematical formulae (page 8/12 comment in first draft) - I did not mean that you put it in a frame. I meant that you should provide more formulae throughout the paper and comments.
Response 3 : According as your comment, we revised the related parts, and added some accounts necessary for explaining the formularized conditions on the case of free-riding with no dead weight loss in public goods supply.
Point 4 : Literature review, and overall literature is very dated. How come that there aren't new findings in the last couple of years?
Response 4 : We reflected your comment by adding recent studies and highlighting the distinction between our research and existing arguments.
Point 5 : Discussion is a summary now. Rather, it should be focused on consequences of the findings, policy implications, theory, comparing results with previous findings, etc.
Response 5 : According as your comments, We revised Discussion section (lines 378-463).
Reviewer 2 Report
In the new version of the paper it is visible, that the authors have made efforts to improve their article. However, it is still poorly elaborated regarding the requirements for writing scientific articles.
In a previous review, I wrote that it was necessary to add the main aspects of the article, such as the originality of the research, the main aim of the paper, the purpose of the paper, what gap of knowledge will you try to fill and and how this article is situated in the current state of the issue.
Chapter titled Literature review still is not really a literature review. In the literature review, not just summarize and cite from others' work, but also an evaluation and comparison is needed. The literature review must be based primarily on recently published papers (in the last three or maximum five years) from journals and conference proceedings. In the current version of the article are almost all sources in Literature review and also in Introduction very old, although probably important. Literature review shall not be limited to old works to be obvious the topicality of the solved issue.
Also in current version of the paper, there is still a complete lack of a Discussion section, where the results of this study should be compared with similar studies (these should be added in the Literature review) and would be given the positioning of this study among others dedicated to this issue. Chapter Discussion is rather Results, not really a discussion. In the beginning, you may briefly summarize main results if appropriate, but keep in mind that Discussion is not a summary of the results.
To summarize, the article needs a complete reorganization of chapters. It is necessary to correctly name the chapters in the paper, especially the chapter of theoretical background, results, to add relevant current sources to Literature review and to complete Discussion.
Author Response
Dear reviewer.
We are very grateful of anonymous reviews’ valuable comments to improve our manuscript. We did our best to develop our draft by reflecting the comments as follows. All revisions were written in Green sentences in the revised manuscript. We also requested a professional native speaker to edit our manuscript as suggested by the reviewer. We look forward to hearing from your journal.
Response to Reviewer 2 Comments
Point 1 : In the new version of the paper it is visible, that the authors have made efforts to improve their article. However, it is still poorly elaborated regarding the requirements for writing scientific articles.
In a previous review, I wrote that it was necessary to add the main aspects of the article, such as the originality of the research, the main aim of the paper, the purpose of the paper, what gap of knowledge will you try to fill and how this article is situated in the current state of the issue.
Response 1 : We revised our article and added those contents in introduction, literature review, Discussion section, and Conclusion.
Point 2 : Chapter titled Literature review still is not really a literature review. In the literature review, not just summarize and cite from others' work, but also an evaluation and comparison is needed. The literature review must be based primarily on recently published papers (in the last three or maximum five years) from journals and conference proceedings. In the current version of the article are almost all sources in Literature review and also in Introduction very old, although probably important. Literature review shall not be limited to old works to be obvious the topicality of the solved issue.
Response 2 : We reconstructed the chapters of this article. Especially chapter 2.1.-2.2 moved to 3.1.-3.2. And then 3.1-3.3. relocated to 3.3-3.5.
And we reflected your comment by adding recent studies in 2.2. and highlighting the distinctions between our research and existing arguments in 2.3.
Point 3 : Also in current version of the paper, there is still a complete lack of a Discussion section, where the results of this study should be compared with similar studies (these should be added in the Literature review) and would be given the positioning of this study among others dedicated to this issue. Chapter Discussion is rather Results, not really a discussion. In the beginning, you may briefly summarize main results if appropriate, but keep in mind that Discussion is not a summary of the results.
Response 3 : According as your comments, We revised Discussion section.
Point 4 : To summarize, the article needs a complete reorganization of chapters. It is necessary to correctly name the chapters in the paper, especially the chapter of theoretical background, results, to add relevant current sources to Literature review and to complete Discussion.
Response 4 : We reflected your comments by correcting the structure of chapter 2 and 3, and changing the title of chapter 4.1.
Reviewer 3 Report
Authors need to improve conclusions as now it mainly rewriting of the arguments provided at the article.
Also they need to check the article as there some grammatical errors like: Contribution to the academic literture and futer study
Author Response
Dear reviewer.
We are very grateful of anonymous reviews’ valuable comments to improve our manuscript. We did our best to develop our draft by reflecting the comments as follows. All revisions were written in Green sentences in the revised manuscript. We also requested a professional native speaker to edit our manuscript as suggested by the reviewer. We look forward to hearing from your journal.
Response to Reviewer 3 Comments
Point 1 : Authors need to improve conclusions as now it mainly rewriting of the arguments provided at the article.
Response 1 : We added lines 465-469 and 478-482 to the Conclusion section.
Point 2 : Also they need to check the article as there some grammatical errors like: Contribution to the academic literture and futer study
Response 2 : We corrected the title of 4.3. And our manuscript have proofreaded professional native speaker.
Round 3
Reviewer 1 Report
The quality of the paper has improved throughout the revisions, thus I recommend it for publication.
Reviewer 2 Report
The authors have made significant changes. As it stands, the coherence of the text is better, and the conceptual framework is more explicit.
I recommend publishing.